# Vine-Winery Byproducts as Precious Resource of Natural Antimicrobials: In Vitro Antibacterial and Antibiofilm Activity of Grape Pomace Extracts against Foodborne Pathogens

**DOI:** 10.3390/microorganisms12030437

**Published:** 2024-02-21

**Authors:** Daniela Sateriale, Giuseppina Forgione, Martina Di Rosario, Chiara Pagliuca, Roberta Colicchio, Paola Salvatore, Marina Paolucci, Caterina Pagliarulo

**Affiliations:** 1Department of Science and Technology, University of Sannio, via F. De Sanctis Snc, 82100 Benevento, Italy; sateriale@unisannio.it (D.S.); gforgione@unisannio.it (G.F.); paolucci@unisannio.it (M.P.); 2Department of Molecular Medicine and Medical Biotechnologies, University of Naples Federico II, Via S. Pansini 5, 80131 Naples, Italy; martina.dirosario@unina.it (M.D.R.); chiara.pagliuca@unina.it (C.P.); roberta.colicchio@unina.it (R.C.); psalvato@unina.it (P.S.); 3CEINGE-Biotecnologie Avanzate s.c.ar.l., Via G. Salvatore 486, 80145 Naples, Italy

**Keywords:** grape pomace extracts, ultrasonic-assisted extraction, natural antibacterials, antibacterial synergy, antibiofilm properties, foodborne bacteria

## Abstract

Grape pomace is the main by-product of vine-winery chains. It requires adequate treatment and disposal but is also an economically underused source of bioactive plant secondary metabolites. This study aimed to investigate the antibacterial effects of polyphenolic extracts from Aglianico (*Vitis vinifera* L.) grape pomace. In particular, hydroethanolic extracts obtained via an ultrasonic-assisted extraction technique were selected for antimicrobial tests. The extracts were screened for their antibacterial effects against foodborne pathogens that were both Gram-positive, in the case of *Staphylococcus aureus* and *Bacillus cereus*, and Gram-negative, in the case of *Escherichia coli* and *Salmonella enterica* subsp. *enterica* serovar Typhimurium, showing variable bacteriostatic and bactericidal effects. In addition, our results demonstrated that the tested grape pomace extracts can reduce the inhibitory concentration of standard antibiotics. Interestingly, selected extracts inhibited biofilm development by *S. aureus* and *B. cereus*. Overall, these new insights into the antibacterial properties of grape pomace extracts may represent a relevant step in the design of novel therapeutic tools to tackle foodborne diseases, and in the management of resistant biofilm-related infections.

## 1. Introduction

The wine industry is one of the most important agricultural sectors in the world, with about 80 million tons of grapes produced annually. In 2018, FAOSTAT (Food and Agriculture Organization Corporate Statistical Database) reported 79 million tons of grape production worldwide, with more than 40% of production taking place in Europe [1]. There are approximately 10,000 grapevine cultivars worldwide, with *Vitis vinifera* L. being the most widely cultivated species for wine production. It originates from southern Europe and can be grown in all temperate regions, including Italian Peninsula [2], where the main red grape varieties include Aglianico, Barbera, Lambrusco, Montepulciano, Nebbiolo, Piedirosso and Sangiovese.

The winemaking process inevitably produces a significant volume of residues, which can pose serious hazards if not properly disposed of. Disposal without any type of treatment can have negative environmental impacts, including water pollution, soil degradation and damage to vegetation [3]. Therefore, the development of alternatives to process the large amount of organic and solid waste generated by the wine industry has become one of the biggest challenges for European wine production [4]. Considering the necessary implementation of wastes management in the wine industry, the development of procedures for their valorization should promote a decrease in environmental impacts, while adding value to the wine production chain byproducts.

Interest in valuing and utilizing byproducts generated at various stages of wine production is increasing. Winemaking byproducts are rich in bioactive secondary plant metabolites belonging to different phytochemical groups, including alkaloids, terpenes and polyphenols [5]. In particular, polyphenols, the most abundant bioactive secondary plant metabolites detected in vine-winery byproducts, are of great interest for the scientific community, and can be valorized for a wide range of industrial sectors, such as agri-food.

Nowadays, the food industry continues to face significant challenges in relation to sustainable food production, supply and consumption, and ensuring food safety to protect consumer health. Foodborne diseases have become a major issue for the global community. Foodborne pathogens of particular concern include *Salmonella* spp., enterohemorrhagic *Escherichia coli*, *Listeria monocytogenes*, *Bacillus* spp., *Staphylococcus aureus* and *Clostridium* spp., which cause food safety issues due to the formation of spores, emetic toxins and biofilms [6]. Consequently, there is a greater awareness of the need to implement innovative strategies in food conservation methodologies. In the last decade, synthetic food preservatives’ harmful effects have led to the search for effective alternatives in the world of natural products, particularly from botanical waste [7]. In this perspective, the byproducts of winemaking can be enhanced in the development of food additives, food supplements, animal feed, but also nutraceuticals and sanitizers in the food industry, thanks to their high polyphenol content [7,8].

Growing scientific evidence suggests that polyphenolic molecules have the potential to exert important bioactive properties [9,10], including antimicrobial effects [11]. Several recent studies have shown how polyphenols extracted from different botanical matrices are able to exert significant in vitro antibacterial and antifungal activity in in vitro models and beneficial effects in in vivo systems [12,13,14,15,16,17]. Regarding waste from vine-winery chains, some studies have demonstrated the antibacterial potential of polyphenolic extracts from winemaking byproducts [18,19]. Generally, the phenolic compounds that are responsible for the antimicrobial activity in winemaking byproducts are phenolic acids, flavonoids, tannins and coumarins [2]. These molecules are particularly abundant in grape pomace, the solid organic material that remains after crushing, draining, and pressing grapes. It consists of a mixture of grape stems, skins, and seeds, and represents the main by-product of the winemaking industry [3]. The composition of grape pomace is complex; it contains 30% neutral polysaccharides, 20% pectic acid derivatives, and 15% phenols [5,20,21]. The high polyphenol content, mostly originating from red grapes, is one of the main important characteristics of grape pomace. Resveratrol, tannins, anthocyanins, phenolic acids, and flavanols are the main bioactive compounds in the phenolic fraction [22]. Combination therapy, in which extracts could be used as antibiotic adjuvants, is a potential application for polyphenol-rich extracts from winemaking byproducts [23,24]. Combination therapy has the potential to prevent bacterial resistance to antibiotics by boosting their bacterteriostatic and bactericidal effects, including those against biofilm-forming bacterial strains, consequently influencing the progression of infections that cannot be treated with conventional antibacterial chemotherapy, including those caused by resistant bacteria [25]. In vitro results about their broad-spectrum antibacterial activity make grape pomace extracts an attractive resource for the purpose of developing new antimicrobials. However, there are still knowledge gaps that researchers are working to address. First of all, there is a wide range of variables, such as extraction processes, waste fraction and grape variety, that could influence the major bioactive compounds in grape pomace extracts, thus resulting in differences in their antimicrobial action [24]. Not all cultivars in the Mediterranean area have been analyzed. For example, few studies have tested the biological properties of grape pomace extracts of Aglianico cultivar, a red grape found in abundance in southern Italy, from which tons of waste can be obtained. In addition, there is little scientific evidence about the potential synergistic effects between different grape pomace extracts and conventional antimicrobials. Understanding how these combinations work together can have implications for developing novel therapeutic strategies against resistant bacterial infections such as biofilm-related ones. Research on antibiofilm activity in grape pomace extracts is an area that needs further exploration. Determining the effectiveness of grape pomace extracts against bacterial biofilms could be significant for addressing the growing problem of resistant infections. 

In this context, our study aims to investigate the antibacterial effects of polyphenolic extracts from Aglianico (*Vitis vinifera* L.) grape pomace against most common causative agents of food infections. The specific aim has been to define the antibacterial profile of hydroethanolic extracts obtained by ultrasonic-assisted extraction techniques against foodborne isolates that are both Gram-positive, such as *Staphylococcus aureus* and *Bacillus cereus*, and Gram-negative, such as *Escherichia coli* and *Salmonella enterica* subsp. *enterica* serovar Typhimurium. Grape pomace extracts have been tested individually and in binary combination with standard antibiotics against the food pathogens to highlight synergistic antimicrobial properties. In addition, the selected grape pomace extracts were tested for their antibiofilm properties, with aim of creating new natural antimicrobials using waste from the vine-winery chain to oppose foodborne diseases and better manage the resistance of biofilm-related infections.

## 2. Materials and Methods

### 2.1. Plant Material

Aglianico (*Vitis vinifera* L.) grape pomace, harvested in November 2022 in the Guardia Sanframondi region (South Italy), was kindly provided as a by-product of red vinification from the winemaking process by the local winery “La Guardiense” (Guardia Sanframondi, BN, Italy). The pressed pomace contained 81.2 ± 0.8% moisture content. It was dried in a thermostatic incubator at a temperature of 50 °C for 48 h, reaching a moisture content of 9.3 ± 0.6%. Grape pomace was ground with a blade homogenizer prior to extraction. Pulverized grape pomace samples were stored at −20 °C to avoid enzymatic degradation of the polyphenols until further use.

### 2.2. Ultrasound-Assisted Extraction

Aqueous and hydroethanolic extracts were prepared by a ultrasound-assisted solid–liquid extraction method. In particular, grape pomace powder was mixed with each extraction buffer, reaching a final dry grape pomace/solvent ratio of 1:10. Two extraction buffers were used: 100% distilled water, for preparation of aqueous extracts, and a hydroalcoholic buffer composed of ethanol and distilled water at 50% (*v*/*v*), for preparation of hydroethanolic extracts. Extracts were prepared by continuous stirring of the pulverized solid matrix in the extraction buffer, using a rotating agitator, for 5 min at room temperature (23 ± 2 °C), followed by incubation in a thermostat-controlled water ultrasonic bath (Digital ultrasonic bath Mod. DU-32, ArgoLab, Carpi, MO, Italy). Several experimental conditions were set for the ultrasound-assisted extraction phase. In particular, the extraction temperature, the ultrasound frequency and the incubation time varied between 25 °C and 50 °C, 20 kHz and 40 kHz and 15 min and 30 min, respectively. Sixteen extracts was prepared, eight aqueous and eight hydroalcoholic; they are listed in Table 1 with their identifying acronyms and extraction characteristics. 

Extracts were filtered through single-use vacuum filtration units (Sterilcup”/Steriltop” Filtration System, Merk-Millipore, Darmstadt, Germany), with filtering membranes with a porosity of 0.45 μm, by use of a water vacuum pump. The extracts were stored at 4 °C until use. Prior to microbiological tests, all extracts were additionally filtered through Millex-GS membrane filters (Merk-Millipore, Darmstadt, Germany), with a porosity of 0.22 μm.

### 2.3. Determination of Polyphenols

The amounts of total polyphenols in the grape pomace extracts were determined by the colorimetric assay of Folin–Ciocalteu [26]. The initial step was to mix aliquots of each extract (0.5 mL) and Folin reagent (0.2 mol L^–1^, 2.5 mL). The mixture was kept away from light for 5 min and then supplemented with 7.5% Na_2_CO_3_ (2 mL). Samples were incubated for 2 h in the dark at room temperature, and then the absorbance was determined using a UV/Vis-6715 Jenway spectrophotometer at 760 nm. The total phenolic content was expressed in gallic acid equivalents per sample (mg GAE g^−1^), thanks to the calibration curve (0–200 mg L^−1^) of gallic acid (Sigma-Aldrich Chemie, Steinheim, Germany). The extracts were analyzed in triplicate.

### 2.4. Foodborne Bacteria and Growth Conditions

The antibacterial activity of the selected hydroethanolic grape pomace extract GpHE_6_ (GpHE_6_, grape pomace hydroethanolic extract n°6) was evaluated against both Gram-positive and Gram-negative foodborne bacteria. In particular, the extracts were tested against ATCC (American Type Culture Collection) collection strains, *Staphylococcus aureus* ATCC 25923 and *Escherichia coli* ATCC 25922, generously provided by IZSM (Istituto Zooprofilattico Sperimentale del Mezzogiorno) institute (Portici, NA, Italy), and against *Bacillus cereus* and *Salmonella enterica* subsp. *enterica* serovar Typhimurium meat isolates, denominated *B. cereus* BC3 and *S.* Typhimurium ST1, obtained from samples of minced pig meat (*B. cereus* BC3) and minced chicken meat (*S.* Typhimurium ST1) at the Laboratory of Microbiology of the Department of Science and Technology, University of Sannio. Details about the isolation and identification of the abovementioned food isolates are available in the authors’ previous study, Sateriale et al. (2023) [27].

Foodborne bacteria (*S. aureus* ATCC 25923, *E. coli* ATCC 25922, *B. cereus* BC3 and *S.* Typhimurium ST1) were aerobically grown, at a temperature of 37 °C, on non-selective, selective and chromogenic and differential media. In particular, Luria–Bertani (LB) medium (CONDA, Madrid, Spain), Baird Parker Base agar medium (CONDA, Madrid, Spain), with tellurite egg yolk emulsion (cat. 5129, CONDA, Madrid, Spain), TBX (Tryptone Bile X-Glucuronide) chromogenic agar (CONDA, Madrid, Spain) medium, Bacillus ChromoSelect agar medium (Sigma-Aldrich S.r.l., Milano, Italy), with Polymyxin B Selektiv-Supplement (Cat. No. P9602, Sigma-Aldrich S.r.l., Milano, Italy) and Xylose Lysine Desoxycholate Agar medium (CONDA, Madrid, Spain) were used.

### 2.5. Antibacterial Assays

#### 2.5.1. Agar Well Diffusion Method 

To qualitatively evaluate the in vitro antibacterial effects of the selected hydroethanolic grape pomace extract GpHE_6_ (GpHE_6_, grape pomace hydroethanolic extract n°6) against foodborne bacteria, an in vitro antimicrobial activity assay was performed using the agar well diffusion method, as reported by Perez (1990) [28], with slight modifications. Briefly, bacteria were grown in LB broth until they reached an optical density (O.D.) of 0.5 at a wavelength of 600 nm. Then, an aliquot of microbial suspension (200 μL) was spread on the agar media and 6 mm wells were punched with sterilized glass Pasteur. Then, the wells were filled up with extract aliquots (25, 50 and 100 µL), and with positive and negative controls. In particular, gentamicin (Sigma-Aldrich S.r.l., Milano, Italy) was used as positive control both for *E. coli* ATCC 25922 and *S.* Typhimurium ST1, while vancomycin (Gold-biotechnology, Saint Louis, Missouri, USA) and amoxicillin (Sigma-Aldrich S.r.l., Milano, Italy) were used as positive controls for *S. aureus* ATCC 25923 and *B. cereus* BC3, respectively; extraction buffer was used as negative control. After the plates were incubated at 37 °C for 24 h, the size of inhibition zones around the wells was measured. The mean diameter of the inhibition zones (MDIZ) (expressed in mm) produced by the extract allowed the evaluation of its in vitro antibacterial activities against the selected microorganisms.

#### 2.5.2. Tube Dilution Method

The susceptibility of foodborne bacteria to different concentrations of the selected hydroethanolic grape pomace extract GpHE_6_ (GpHE_6_, grape pomace hydroethanolic extract n°6), also in binary combination with standard antibiotics, was determined by the tube dilution method with broth standard inoculum 1 × 10^5^ CFU/mL (Colonies Forming Units/mL), according to Clinical and Laboratory Standards Institute (CLSI) 2022 guidelines [29]. A quantitative evaluation of the antibacterial effects of extract and extract–antibiotic binary combinations could be achieved by determining minimum inhibitory concentration (MIC) and minimum bactericide concentration (MBC) values for each tested antibacterial agent. 

Bacterial cultures were incubated in aerobic conditions at 37 °C for 24 h, with extract at increasing concentrations (0, 2.5, 5, 10, 15, 20, 25, 30, 35, 40, 45, 50, 55 and 60 mg mL^−1^), ensuring a constant agitation. Gentamicin (concentration range, 1–20 µg mL^−1^) was used as positive control for both *E. coli* ATCC 25922 and *S.* Typhimurium ST1, while vancomycin (concentration range, 0.5–5 µg mL^−1^) and amoxicillin (concentration range, 1–10 µg mL^−1^) were used as positive controls for *S. aureus* ATCC 25923 and *B. cereus* BC3, respectively. Extract–antibiotic binary combination (1:1 ratio) between GpHE_6_ and standard antibiotics were also tested at increasing concentrations. The hydroalcoholic extraction buffer, composed of ethanol and distilled water at 50% (*v*/*v*), was used as negative control, by adding increasing volumes on the basis of relative tested concentrations of extract and extract–antibiotic binary combinations.

After incubation, MIC values were determined by evaluating tube turbidity. Subsequently, serial dilutions of bacterial suspensions were prepared and aliquots were spread on LB agar plates. After a second incubation at 37 °C for 24 h, a vital bacterial count was performed for MBC determination. In particular, MIC was assigned to the lowest concentration of each in vitro antibacterial agent capable of preventing bacterial growth, while MBC was assigned to the lowest concentration of each in vitro antibacterial agent able to kill 99% of bacteria from the initial inoculum. 

### 2.6. In Vitro Synergy Analysis with Standard Antibiotics 

The susceptibility of foodborne bacteria to different concentrations of the binary combinations (1:1 ratio) between the selected hydroethanolic grape pomace extract and standard antibiotics was determined by the tube dilution method, with broth standard inoculum 1 × 10^5^ CFU/mL (Colonies Forming Units/mL), according to Clinical and Laboratory Standards Institute (CLSI) 2022 guidelines [29]. In particular, the effects of the selected hydroethanolic grape pomace extract GpHE_6_ (GpHE_6_, grape pomace hydroethanolic extract n°6) and standard antibiotics (amoxicillin, gentamicin and vancomycin) were deemed partially synergistic, synergistic, indifferent or antagonistic against selected foodborne bacteria, through the measuring of the fractional inhibitory concentration index (FICI) of their binary combination, similarly to Sateriale et al. (2020) [30]. The formulas listed below were utilized: Fractional Inhibitory Concentration (FIC) = MIC of antimicrobial agent in the binary combination/MIC of single antimicrobial agent; FICI = FIC of antimicrobial agent 1 + FIC of antimicrobial agent 2. In accordance with the Odds’ interpretation [31], the FIC was characterized as the minimum inhibiting concentration (MIC) of the antimicrobial agent used in combination, compared to the MIC of the same antimicrobial agent used alone. The sum of the FICs obtained for the binary combination was used to define FICI, which represents the type of interaction between the different antibacterial agents against each foodborne bacteria (FICI ≤ 0.5, synergy; 0.5 < FICI ≤ 1, partial synergy; 1 < FICI ≤ 4, indifference; FICI > 4, antagonism).

### 2.7. Antibiofilm Assays

#### 2.7.1. Tissue Culture Plate Method

The so-called tissue culture plate method (TCPM) was carried out to measure the biofilm biomass of *S. aureus* ATCC 25923 and *B. cereus* BC3, as described by Sateriale et al. (2020) [32]. After growing overnight at 37 °C for 24 h in LB broth in aerobic incubation cultures of selected microorganisms, the OD_600nm_ of fresh cultures were further adjusted until 0.5 OD. Then, 96-well microplates (Nunc™, Thermo Scientific, Roskilde, Denmark) were used to dispense bacteria cultures (aliquots of 200 µL). Sterile LB broth was used as negative control. After incubation at 37 °C for 24 h without shaking, allowing bacterial adhesion, the not-adherent bacterial cells were washed away. In particular, three washings with phosphate-buffered saline (1 × PBS, pH 7.3) were carried out, before adherent biofilms were fixed with 85% ethanol (Sigma-Aldrich, Merck KGaA, Darmstadt, Germany) for 15 min. Finally, fixed biofilms were stained with 0.2% crystal violet (Sigma-Aldrich, Merck KGaA, Darmstadt, Germany) for 5 min. To remove any remaining stains, the plates were washed with deionized water and dried upside down in a thermostat at 30 °C for 10 min. Then, 85% ethanol (Sigma-Aldrich, Merck KGaA, Darmstadt, Germany) was added and the optical density of bacterial cultures (OD_bc_) was recorded by a microplate reader (Bio-Rad Microplate reader, Model 680) at 600 nm. Comparing the values measured for sterile LB broth used as the negative control (OD_nc_), the tested foodborne bacteria were classified as non-adherent (OD_bc_ ≤ O_nc_), weakly adherent (OD_nc_ < OD_bc_ ≤ 2 OD_nc_), moderately adherent (2 OD_nc_ < OD_bc_ ≤ 4 OD_nc_) and strongly adherent (4 OD_nc_ < OD_bc_).

#### 2.7.2. Biofilm Formation Inhibition Assay

To evaluate the ability of the selected hydroethanolic grape pomace extract GpHE_6_ (GpHE_6_, grape pomace hydroethanolic extract n°6) to inhibit biofilm formation by *S. aureus* ATCC 25923 and *B. cereus* BC3, minor changes to the tissue culture plate method were performed. Aliquots of extract were added to bacterial cultures in 96-well microplate wells, reaching increasing final concentrations (0, 10, 20, 40, 80, 100 mg mL^−1^). Sterile LB broth was used as negative control. After aerobic incubation in appropriate growth conditions, the staining step of the TCPM method with crystal violet was performed. Finally, microplate reading at the wavelength of 600 nm was carried out. Results were reported as the percentage of biofilm formation inhibition (BII %), similarly to Bakkiyaraj et al. (2013) [33] as follows:
$$\mathrm{Percentage}\mathrm{index}\mathrm{of}\mathrm{biofilm}\mathrm{inhibition}\;(\mathrm{BII}\%)=\left(\frac{{{OD}_{control}-OD}_{assay}}{{OD}_{control}}\right)\times 100$$
where OD_control_ corresponds to the mean optical density measured for bacterial biofilms grown in the absence of extract, while OD_assay_ is the mean optical density measured for bacterial biofilms grown in the presence of extract. The minimum biofilm inhibition concentration (MBIC) was defined as the lowest concentration of antibacterial agent capable of producing bacterial biofilm inhibition.

### 2.8. Statistical Data Analysis

All experiments were performed in triplicate, with independent bacterial cultures for antimicrobial assays. The results obtained were analyzed and graphically reported by using “GraphPad Prism 8.00” software, validating the statistical significance by the one-way ANOVA test, with Dunnett’s and Tukey’s post hoc tests. In all cases, *p* values < 0.05 were considered statistically significant. 

## 3. Results 

### 3.1. Extraction Yield from Aglianico Grape Pomace and Major Polyphenolic Constituents

The extraction yield of polyphenolic extracts prepared from red grape pomace (*V. vinifera* L., Aglianico cultivar) by an ultrasound-assisted solid–liquid extraction method varied considering the analyzed extracts. Different values of total phenolic content (TPC) were obtained depending on the extraction temperature, the ultrasound frequency, the extraction time and the solvent used for the extraction. The obtained concentrations of total polyphenols in the tested grape pomace polyphenolic extracts are reported in Figure 1, indicated as mg of gallic acid equivalents (GAE) for g of dry grape pomace sample. The higher concentration in total polyphenols was registered in GpHE_6_ and GpHE_7_ extracts, followed by GpHE_5_ and GpHE_8_ extracts, while the lowest levels of TPC were detected in aqueous extracts GpAE_1_, GpAE_2_, GpAE_3_ and GpAE_4_. Appendix A shows the total polyphenols content of aqueous and hydroethanolic extracts of Aglianico grape pomace obtained by ultrasonic-assisted extraction.

### 3.2. In Vitro Antibacterial Activity of Hydroethanolic Grape Pomace Extract against Gram-Positive and Gram-Negative Foodborne Bacteria

The hydroethanolic grape pomace extract GpHE_6_, selected for the higher extraction yield, exhibited an appreciable inhibitory activity against tested Gram-positive foodborne bacteria, i.e., *Staphylococcus aureus* and *Bacillus cereus*, as demonstrated by the evident inhibition zones of bacterial growth estimated through the agar well diffusion method (Appendix A). Figure 2 shows the mean diameters of the inhibition zones (MDIZ) of bacterial growth exerted by the tested extract against *S. aureus* ATCC 25923 strain (Figure 2A) and *B. cereus* BC3 isolate (Figure 2B). The MDIZ values are reported in detail in Appendix A. Regarding the Gram-negative bacteria, *Escherichia coli* ATCC 25922 were shown to be less sensitive to hydroethanolic grape pomace extract (Figure 2C and Appendix A), while *Salmonella enterica* subsp. *enterica* serovar Typhimurium ST1 food isolate showed resistance to the extract (Figure 2D and Appendix A).

Vancomycin, amoxicillin and gentamicin, used as positive controls, showed antibacterial efficacy against tested bacteria; no effects were observed for the negative control. 

The in vitro antibacterial activity of hydroethanolic grape pomace extract was also confirmed by quantitative assays. The minimum inhibitory concentration (MIC) and minimum bactericidal concentration (MBC) values are reported in Table 2. The hydroethanolic grape pomace extract showed bacteriostatic and bactericidal effects, with a dose-dependent trend, against *S. aureus* ATCC 25923, *B. cereus* BC3 and *E. coli* ATCC 25922 foodborne bacteria. The inhibitory effect of extract on the mentioned food pathogens was appreciable both when they were used individually and in binary combination. The hydroethanolic grape pomace extract showed a bacteriostatic effect against *S.* Typhimurium ST1, with a MIC value of 60 mg mL^−1^, but no bactericidal effect was detected at tested concentrations for the *Salmonella* food isolate.

### 3.3. Synergistic Inhibitory Effect of Binary Combination of Hydroethanolic Grape Pomace Extract and Standard Antibiotics against Gram-Positive and Gram-Negative Foodborne Bacteria

The in vitro antibacterial activity of the binary combination (1:1 ratio) of hydroethanolic grape pomace extract GpHE_6_ (GpHE_6_, grape pomace hydroethanolic extract n°6) and standard antibiotics (VNC, vancomycin; AMX, amoxicillin; GNT, gentamicin) was evaluated by the determination of fractional inhibitory concentration (FIC) value, for each antibacterial agent, and of the FIC index (FICI) for each binary combination. FIC and FICI values are reported in Table 3. 

The binary combinations GpHE_6_-VNC and GpHE_6_-AMX showed partially synergistic and synergistic in vitro antibacterial effects against *S. aureus* ATCC 25923 and *B. cereus* BC3, respectively, as shown by the FIC index (FICI) values (Table 3). The other tested binary combination (GpHE_6_-GNT) showed indifference against selected Gram-negative foodborne bacteria, i.e., *E. coli* ATCC 25922 and *S.* Typhimurium ST1.

### 3.4. In Vitro Antibiofilm Activity of Hydroethanolic Grape Pomace Extract against Staphylococcus aureus and Bacillus cereus Foodborne Bacteria

The antibiofilm activity of hydroethanolic grape pomace extract GpHE_6_ was evaluated against *Staphylococcus aureus* ATCC 25923 and *Bacillus cereus* BC3 foodborne bacteria by the tissue culture plate method to assess the impact of this natural agent on biofilm formation and maturation in the food industry. 

Figure 3 and Figure 4 show how GpHE_6_ extract was able to significantly inhibit the biofilm development of both *S. aureus* ATCC 25923 and *B. cereus* BC3, respectively. Measurements of optical density (OD_600nm_) values were performed for biofilms grown in the absence and in the presence of increasing concentrations of GpHE_6_ (10, 20, 40, 80, 100 mg mL^−1^). The adhesion level for each test condition was carried out thanks to the comparison with OD values of bacterial cultures with the negative control values, represented by sterile broth medium. This allowed the bacterial isolates to be classified as non-adherent, weakly adherent, moderately adherent and strongly adherent.

In detail, *S. aureus* ATCC 25923 showed moderate adherence (2 OD_nc_ < OD_bc_ ≤ 4 OD_nc_) both in the absence of GpHE_6_ (0 mg mL^−1^) and at a concentration of 10 mg mL^−1^. A gradual decrease in adhesion of *S. aureus* strain was recorded in the presence of increasing concentrations of extract, reaching the absence of adherence at the concentration of 80 mg mL^−1^ of GpHE_6_ (OD_bc_ ≤ O_nc_) (Figure 3). Therefore, the concentration of 80 mg mL^−1^ can be indicated as the MBIC for GpHE_6_ against *S. aureus* ATCC 25923.

*B. cereus* BC3 food isolate showed moderate adherence (2 OD_nc_ < OD_bc_ ≤ 4 OD_nc_) in the absence of GpHE_6_ (OD_bc_ ≤ O_nc_), while a decrease in adhesion was observed in the presence of increasing concentrations of extract, until a non-adherent level was reached at the concentration of 100 mg mL^−1^ of GpHE_6_ (OD_bc_ ≤ O_nc_) (Figure 4), identified as the minimum biofilm inhibition concentration (MBIC) value for the tested extract against *B. cereus* meat isolate.

At increasing concentrations of GpHE_6_, a progressive increase in the percentage of biofilm inhibition was observed for both *S. aureus* (Figure 5A) and *B. cereus* (Figure 5B) foodborne bacteria. In particular, GpHE_6_ was able to inhibit biofilm formation by *S. aureus* ATCC 25923 and *B. cereus* BC3, reaching inhibition percentages of 79.93 ± 2.40% (Figure 5A) and 83.36 ± 1.99% (Figure 5B), respectively, at the maximum tested concentration of 100 mg mL^−1^.

## 4. Discussion

There is an increased interest in the valorization and use of byproducts generated at different stages of wine production. In particular, complex polyphenolic mixtures can be extracted from grape pomace and exploited for their antimicrobial properties. However, given that numerous factors can influence the polyphenolic composition and, consequently, the antimicrobial activity of these botanical extracts, further detailed scientific evidence is needed to demonstrate the antibacterial efficacy of polyphenolic extracts from byproducts of the wine industry. 

Although several studies investigated polyphenols in grape pomace extracts and their biological properties, this study represents one of the very few contributions in the literature that concerns the extractive yield of polyphenols from grape pomace of Aglianico (*Vitis vinifera* L.) cultivar in function of different chemical–physical parameters. This study reports a good extractive yield by assisted ultrasonic extraction for total polyphenol content in aqueous and hydroethanolic extracts prepared from dry Aglianico grape pomace samples. High extraction yields may be explained in terms of propagation of ultrasound pressure waves and their resulting effects. Cavitation forces generated by ultrasonic waves result in a localized pressure that breaks down plant tissues and enhances the release of intracellular substances into the solvent [34]. The extraction yields significantly improved as the extraction temperature increased from 25 to 50 °C. The increased solubility and diffusion coefficients of the extracted compounds and the decreased solvent viscosity at high temperatures could be the cause of this effect [35,36]. In addition, at higher temperatures, the vapor pressure is higher and more bubbles are created, thus enhancing the collapsing process of plant cell walls [35]. The extraction yield improved with increased ultrasound frequency from 20 kHz to 40 kHz. This trend may be due to the improved cavitation and mechanical effects of ultrasounds increasing the contact surface area between solid and liquid surfaces and causing greater penetration of the solvent into the pomace matrix [36]. However, excessive levels of ultrasound frequency during extraction processes could lead to a decrease in yield, possibly due to degradation of the plant material [37]. In our experiments, the extraction yield was also time-dependent, but with minor variations, for ultrasound times extended from 15 to 30 min. A similar result during ultrasonic-assisted extraction was reported by other studies and was attributed to the fact that ultrasonic waves affect the rate of mass transfer mainly in the phase of solvent penetration [35]. Thus, the efficient extraction time for achieving a good yield of grape pomace phenolics could be about 10–20 min [36]. The extraction solvent caused the most significant changes in extraction yield. In particular, the hydroethanolic extracts showed significantly higher total polyphenolic content than the aqueous ones, showing values higher than 50 mg GAE g^−1^ of dry grape pomace. Several studies reported that the phenolics solubility may be improved by hydroalcoholic extractants, which can favor isolation of polar and mid-polar compounds [38,39]. According to Drevelegka & Goula (2020), for instance, mixtures of ethanol and water are more efficient in extracting phenolic compounds compared to mono-component solvent systems [40]. Considering the results obtained from this preliminary screening and the close relationship between the amount of total polyphenols in natural extracts and their antimicrobial activity, the hydroethanolic grape pomace extract prepared at T of 50 °C, with ultrasonic frequency of 40 kHz, for incubation time of 15 min (GpHE_6_) was selected for antimicrobial, synergistic and antibiofilm analysis. 

Several in vitro antimicrobial assays allowed us to verify the ability of selected hydroethanolic grape pomace extract GpHE_6_ to effectively counteract the growth of four important foodborne pathogenic bacteria, both Gram-positive, *Staphylococcus aureus* and *Bacillus cereus*, and Gram-negative, *Escherichia coli* and *Salmonella enterica* subsp. *enterica* serovar Typhimurium. First, GpHE_6_ in vitro antimicrobial activity was evaluated using the agar well-diffusion method. From this preliminary screening, it was determined that the tested grape pomace extract showed the highest antibacterial activity against *S. aureus* and *B. cereus*, comparable to that of antibiotics tested as positive control. Other studies reported similar results, confirming the antibacterial potential of extracts from winemaking byproducts [24,41]. Similarly, recent studies confirmed the significant antibacterial activity of grape pomace extracts from Muscadine and Cabernet Sauvignon against Gram-positive bacteria [42,43,44], due to high concentrations of active compounds, such as phenolic acids, quinones, saponins, flavonoids, tannins, coumarins, terpenoids, and alkaloids [2]. However, only a few studies demonstrated the antibacterial activity of Aglianico grape pomace extracts [45] against pathogenic bacteria, thus confirming the added value of this study. The hydroethanolic grape pomace extract tested in this study showed moderate inhibitory activity against *E. coli* ATCC 25922 strain, while no effect was observed against *S.* Typhimurium ST1 food isolate. These results are in accordance with literature studies, showing the higher grape pomace antibacterial activity against Gram-positive bacteria compared to Gram-negative ones [5,46]. In vitro antibacterial activity was also confirmed by quantitative antimicrobial assays. The selected hydroethanolic grape pomace extract (GpHE_6_) showed bacteriostatic effects against the four tested bacteria but showed bactericidal effects only against the two Gram-positive foodborne bacteria (*S. aureus* ATCC 25923 and *B. cereus* BC3) and against *E. coli* ATCC 25922. The MIC and MBC values ranged between 15 mg mL^−1^ (GpHE_6_ vs. *S. aureus* ATCC 25923) and 60 mg mL^−1^ (GpHE_6_ vs. *S.* Typhimurium ST1), and between 40 mg mL^−1^ (GpHE_6_ vs. *S. aureus* ATCC 25923) and 60 mg mL^−1^ (GpHE_6_ vs. *E. coli* ATCC 25922), respectively, in agreement with the literature data. In several studies, it was found that Gram-positive bacteria had a wider inhibitory spectrum and lower MIC values than Gram-negative bacteria [46]. Grape pomace extracts are known to have bactericidal properties [47], with widely variable MBC values, reflecting both bacterial sensitivity and concentration of specific antibacterial phenols in the extracts [5], but knowledge about the bactericidal power of extracts of Aglianico grape pomace was very low until this study.

Particularly significant are the results regarding synergistic activity of hydroethanolic grape pomace extract with antibiotics commonly used in therapy (VNC—vancomycin; AMX—amoxicillin). The binary combinations GpHE_6_-VNC and GpHE_6_-AMX showed partial synergistic and synergistic in vitro antibacterial effects against *S. aureus* ATCC 25923 and *B. cereus* BC3, respectively. A synergism between grape pomace extracts and antibiotics belonging to different classes has been demonstrated [48], but this work demonstrated, for the first time, the adjuvant properties of Aglianico grape pomace extracts with standard antibiotics. Among possible mechanisms of action of synergistic antibacterial effects between grape pomace extracts and antibiotics, modification of the active sites in bacterial cells, increased membrane permeability and inhibition of bacterial enzymes involved in antibiotic modification are among the main processes that seem to be involved in the synergistic properties of polyphenol-rich extracts from winemaking byproducts [49]. However, knowledge on the mechanisms of plant polyphenols–antibiotic synergism is still incomplete. This constitutes an important bottleneck and highlights the need for further research. In this contest, these results are of great interest and suggest the potential use of grape pomace extracts in combination therapy as antibiotic adjuvants.

Finally, this study also investigated, for the first time, the antibiofilm activity of hydroethanolic grape pomace extract from the Aglianico cultivar. In detail, the tested extract was able to inhibit biofilm formation of *Staphylococcus aureus* ATCC 25923 and *Bacillus cereus* BC3 foodborne bacteria, in a dose-dependent way. Foodborne strains can increase their physical and chemical resistance by developing biofilms on different surfaces, which enhances their persistence in several environments [50]. As a consequence, the obtained results are extremely significant. Even if these results are in agreement with other studies about the antibiofilm effects of grape pomace extracts [43], most of the recent studies tested extracts obtained from vine leaves and grape pomace of other varieties of red grapes than Aglianico [45]. The antibiofilm properties of Aglianico grape pomace extracts represent the further novelty of this study. Proven antibiofilm effects depend closely on the composition of extracts. A link between the phenolic content of extracts and their antibacterial and antibiofilm activity has been established in some studies [41,43]. The antimicrobial activity of phenolic extracts was generally higher than pure phenolic molecules, suggesting a potential synergistic activity between the constituent compounds of the extracts [41]. The antibacterial activity of hydroethanolic grape pomace extract tested in this study could be related to the presence of flavonoids (catechin, epicatechin, trans-resveratrol and quercetin) and anthocyanins (malvidin derivatives), in agreement with literature data [51,52,53,54,55], while phenolic acids (gallic acid and caffeic acid) would play an important role in reducing biofilm formation [5].

In conclusion, the encouraging results regarding the demonstrated ability of Aglianico (*V. vinifera* L.) grape pomace extract to exert strong antibacterial activity, to synergistically work with antibiotics and to inhibit biofilm formation, provide evidence that they can be considered valid candidates for the development of natural antimicrobial agents to control foodborne pathogens. From this perspective, pomace extracts from Aglianico red grape could represent a precious source of abundant and effective antimicrobials for several applications, such as innovative antibiotic therapy and food safety based on vegetable processing waste.

## Figures and Tables

**Figure 1 microorganisms-12-00437-f001:**
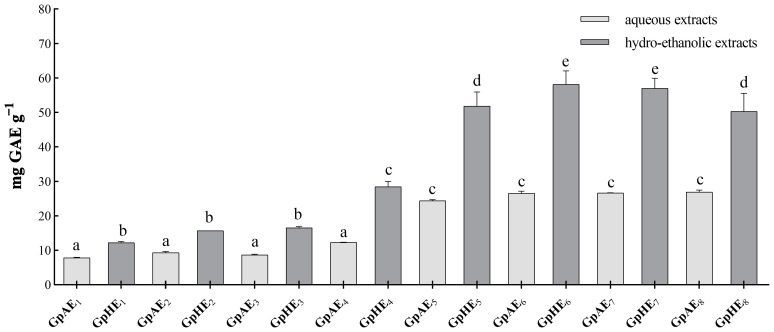
Total polyphenols content (TPC) of aqueous and hydroethanolic extracts of Aglianico (*V. vinifera* L.) grape pomace obtained by ultrasonic-assisted extraction. The results estimated by Folin–Ciocalteu assay are expressed in mg gallic acid (GAE) equivalents per g of dry solid matrix. Results are reported as mean standard deviations of data obtained from tripled experiments. One-way ANOVA test was performed to evaluate statistical significance. Tukey’s post hoc test (*p* < 0.05) allowed us to examine the statistical significance for multiple comparisons between bars. Different letters (a–e) indicate significant differences between bars; bars with no significant differences receive the same letter. GpAE—grape pomace aqueous extract; GpHE—grape pomace hydroethanolic extract. Details about extraction conditions are in Table 1.

**Figure 2 microorganisms-12-00437-f002:**
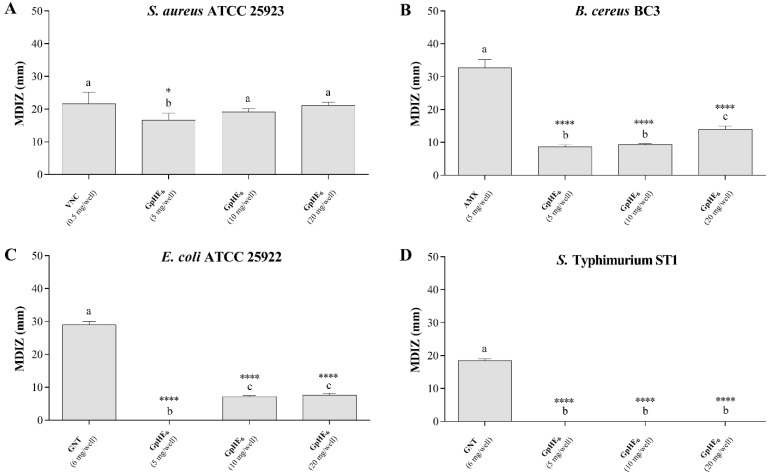
In vitro antibacterial activity of hydroethanolic grape pomace extract against *Staphylococcus aureus* ATCC 25923 (**A**), *Bacillus cereus* BC3 (**B**), *Escherichia coli* ATCC 25922 (**C**) *Salmonella enterica* subsp. *enterica* serovar Typhimurium ST1 (**D**) foodborne bacteria. Results were obtained by agar well diffusion method in triplicate assays with independent cultures. The mean diameters of inhibition zone, reported as mean values ± standard deviation (expressed in mm), are graphically represented. One-way ANOVA test was performed to evaluate statistical significance. Bars comparison with positive control bar (absence of extract) was analyzed by Dunnett’s post hoc test (*p* < 0.05), using asterisks to indicate statistical significance respect to the positive control (**** *p* < 0.0001; * *p* < 0.05). Tukey’s post hoc test (*p* < 0.05) allowed us to examine the statistical significance for multiple comparisons between bars. Different letters (a–c) indicate significant differences between bars; bars with no significant differences receive the same letter. MDIZ—mean diameter of the inhibition zone; GpHE_6_—grape pomace hydroethanolic extract n°6; VNC—vancomycin; AMX—amoxicillin; GNT—gentamicin.

**Figure 3 microorganisms-12-00437-f003:**
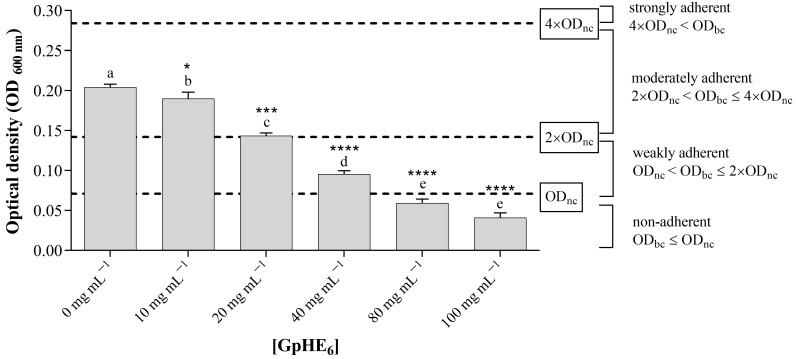
Adherence levels of *Staphylococcus aureus* ATCC 25923 strain. Optical density (OD) values, detected by absorbance reading at a wavelength of 600 nm with a microplate reader, of bacterial biofilms developed by *S. aureus* ATCC 25923 in the absence and in the presence of increasing concentrations (10, 20, 40, 80, 100 mg mL^−1^) of GpHE_6_. The comparison with negative control, represented by the broth medium, allowed us to determine the level of adherence for each experimental condition. One-way ANOVA test was performed to evaluate statistical significance. Bars comparison with positive control bar (absence of GpHE_6_) was analyzed by Dunnett’s post hoc test (*p* < 0.05), using asterisks to indicate statistical significance with respect to the positive control (* *p* < 0.05; *** *p* < 0.001; **** *p* < 0.0001). Tukey’s post hoc test (*p* < 0.05) allowed us to examine the statistical significance for multiple comparisons between bars. Different letters (a–e) indicate significant differences between bars; bars with no significant differences receive the same letter. GpHE_6_—grape pomace hydroethanolic extract n°12; OD_bc_—bacterial culture optical density; OD_nc_—negative control optical density.

**Figure 4 microorganisms-12-00437-f004:**
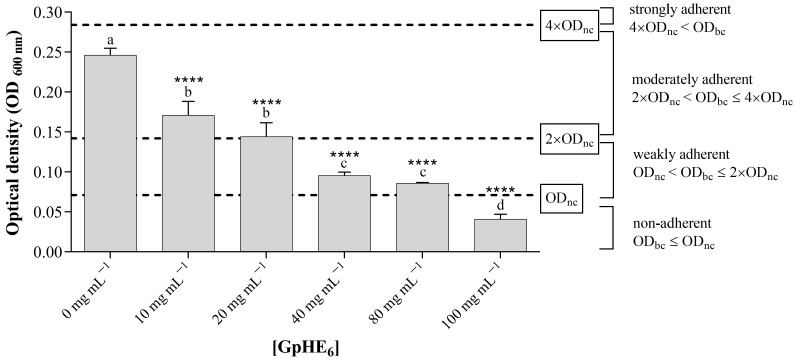
Adherence levels of *Bacillus cereus* BC3 food isolate. Optical density (OD) values, detected by absorbance reading at a wavelength of 600 nm with a microplate reader, of bacterial biofilms developed by *B. cereus* BC3 in the absence and in the presence of increasing concentrations (10, 20, 40, 80, 100 mg mL^−1^) of GpHE_6_. The comparison with negative control, represented by the broth medium, allowed us to determine the level of adherence for each experimental condition. One-way ANOVA test was performed to evaluate statistical significance. Bars comparison with positive control bar (absence of GpHE_6_) was analyzed by Dunnett’s post hoc test (*p* < 0.05), using asterisks to indicate statistical significance with respect to the positive control (**** *p* < 0.0001). Tukey’s post hoc test (*p* < 0.05) allowed us to examine the statistical significance for multiple comparisons between bars. Different letters (a–d) indicate significant differences between bars; bars with no significant differences receive the same letter. GpHE_6_—grape pomace hydroethanolic extract n°6; OD_bc_—bacterial culture optical density—OD_nc_—negative control optical density.

**Figure 5 microorganisms-12-00437-f005:**
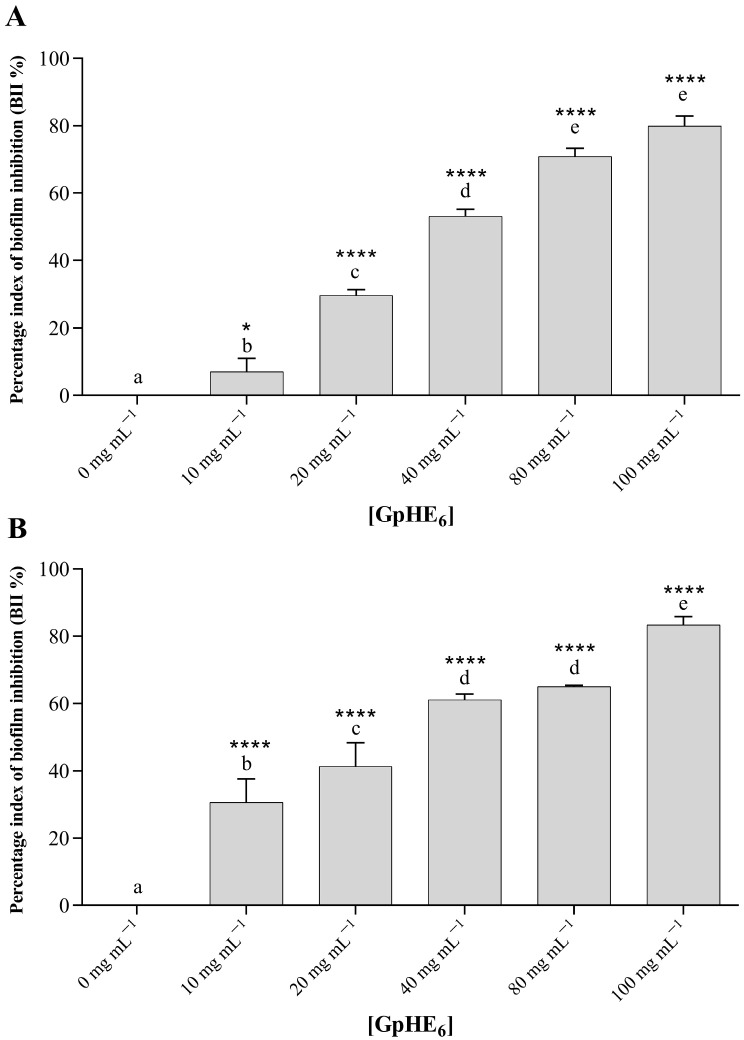
Inhibitory effect of hydroethanolic grape pomace extract against biofilm formation by *Staphylococcus aureus* ATCC 25923 (**A**) and *Bacillus cereus* BC3 (**B**) foodborne bacteria. Graphs show the percentage inhibition values of biofilms in the presence and absence of increasing concentrations of GpHE_6_ (10, 20, 40, 80, 100 mg mL^−1^). One-way ANOVA test was performed to evaluate statistical significance. Bars comparison with positive control bar (absence of GpHE_6_) was analyzed by Dunnett’s post hoc test (*p* < 0.05), using asterisks to indicate statistical significance with respect to the positive control (* *p* < 0.05; **** *p* < 0.0001). Tukey’s post hoc test (*p* < 0.05) allowed us to examine the statistical significance for multiple comparisons between bars. Different letters (a–e) indicate significant differences between bars; bars with no significant differences receive the same letter. GpHE_6_—grape pomace hydroethanolic extract n°6.

**Table 1 microorganisms-12-00437-t001:** Ultrasonic-assisted extraction conditions for preparing aqueous and hydroethanolic polyphenolic extracts from Aglianico (*Vitis vinifera* L.) grape pomace.

N°	Plant Material	Extraction Buffer	Grape Pomace/ Solvent Ratio	Extraction Temperature	Extraction Time	Ultrasound Frequency	Extract Acromyn
1	Dry grape pomace of Aglianico (*V. vinifera* L.)	100% distilled water	1:10	25 °C	15 min	20 kHz	GpAE_1_
2				25 °C	15 min	40 kHz	GpAE_2_
3				25 °C	30 min	20 kHz	GpAE_3_
4				25 °C	30 min	40 kHz	GpAE_4_
5				50 °C	15 min	20 kHz	GpAE_5_
6				50 °C	15 min	40 kHz	GpAE_6_
7				50 °C	30 min	20 kHz	GpAE_7_
8				50 °C	30 min	40 kHz	GpAE_8_
9		Ethanol and distilled water at 50% (*v*/*v*)		25 °C	15 min	20 kHz	GpHE_1_
10				25 °C	15 min	40 kHz	GpHE_2_
11				25 °C	30 min	20 kHz	GpHE_3_
12				25 °C	30 min	40 kHz	GpHE_4_
13				50 °C	15 min	20 kHz	GpHE_5_
14				50 °C	15 min	40 kHz	GpHE_6_
15				50 °C	30 min	20 kHz	GpHE_7_
16				50 °C	30 min	40 kHz	GpHE_8_

GpAE—grape pomace aqueous extract; GpHE—grape pomace hydroethanolic extract.

**Table 2 microorganisms-12-00437-t002:** Quantitative evaluation of in vitro antibacterial activity of hydroethanolic grape pomace extract, used individually and in binary combination with standard antibiotics, against foodborne bacteria.

Antibacterial Agent	*S. aureus* ATCC 25923		*B. cereus* BC3		*E. coli* ATCC 25922		*S. Typhimurium* ST1	
	MIC (mg mL^−1^)	MBC (mg mL^−1^)	MIC (mg mL^−1^)	MBC (mg mL^−1^)	MIC (mg mL^−1^)	MBC (mg mL^−1^)	MIC (mg mL^−1^)	MBC (mg mL^−1^)
GpHE_6_	15	40	20	50	40	60	60	nd
VNC	0.0015	0.0025	nt	nt	nt	nt	nt	nt
AMX	Nt	nt	0.003	0.005	nt	nt	nt	nt
GNT	Nt	nt	nt	nt	0.004	0.01	0.005	0.02
GpHE_6_+VNC (1:1 ratio)	0.001	0.002	nt	nt	nt	nt	nt	nt
GpHE_6_+AMX (1:1 ratio)	Nt	nt	0.001	0.005	nt	nt	nt	nt
GpHE_6_+GNT (1:1 ratio)	Nt	nt	nt	nt	0.004	0.01	0.005	0.02

MIC—minimum inhibitory concentration; MBC—minimum bactericidal concentration; GpHE_6_—grape pomace hydroethanolic extract n°6; VNC—vancomycin; AMX—amoxicillin; GNT—gentamicin; nd—not detected; nt—not tested.

**Table 3 microorganisms-12-00437-t003:** Synergistic antibacterial effects of binary combinations of hydroethanolic grape pomace extract with standard antibiotics against foodborne bacteria.

Bacterial Isolate	Binary Combinations	Individual FIC	FIC Index (FICI)	Interaction Interpretation
*S. aureus* ATCC 25923	GpHE_6_+VNC	0.0001–0.6667	0.6668	partial synergy
*B. cereus* BC3	GpHE_6_+AMX	0.0001–0.3333	0.3334	synergy
*E. coli* ATCC 25922	GpHE_6_+GNT	0.0001–1.0000	1.0001	indifference
*S.* Typhimurium ST1	GpHE_6_+GNT	0.0001–1.1000	1.1000	indifference

GpHE_6_—grape pomace hydroethanolic extract n°6; VNC—vancomycin; AMX—amoxicillin; GNT—gentamicin; FIC—fractional inhibitory concentration; FICI—fractional inhibitory concentration index; FICI ≤ 0.5—synergy; 0.5 < FICI ≤ 1—partial synergy; 1 < FICI ≤ 4—indifference; FICI > 4—antagonism.

## Data Availability

Data are contained within the article and Appendix A.

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
