# Peer review of "Vine-Winery Byproducts as Precious Resource of Natural Antimicrobials: In Vitro Antibacterial and Antibiofilm Activity of Grape Pomace Extracts against Foodborne Pathogens"

_microorganisms, 2024, doi:10.3390/microorganisms12030437_

Round 1

Reviewer 1 Report

Comments and Suggestions for Authors

The manuscript is interesting and fit with the scope of the journal. It contains however several drawbacks and flaws that have to be considered.

It is not evident what is the new scientific contribution of the present work. If it is mainly the variety of the red grape that otherwise give the same antimicrobial effects as other varieties, the results are too weak to be presented as an article. Have the effects of the chosen bacteria been published previously, or is anything here new? Nothing with respect to methodology nor chemical analysis is new. To me it seems like the overall new contribution of knowledge is low.

Abstract: Erase first sentence. The same content follows in the next sentence. Also erase last half of the sentence in line 23-24.

Introduction: Good intro, but it should also contain what is not yet known (the knowledge gaps) and how this work contributes to close/fill these gaps.

Materials and methods:

Line 115-: Unclear whether the extract is frozen-thawed-frozen in this process.

Line 120-: It is unclear at this stage whether you talk about one or two extracts that are treated with water and water-ethanol, respectively. There is nothing said about the mass of dried, pulverized pomace that was used for extraction.

Line 131-160: Arrange all this into a table. That would give a better overview.

Line 175: Rather “The extracts were analyzed in triplicate.”

Line 176-188+appendix: The chemical analysis has a very low standard, and the identifications (only qualitative) are not reliable. There is nothing news here anyway, and the “results” are not used elsewhere in the work. If not improved and used, leave everything out. That includes lines 341-347.

Line 213-: With respect to the HE-extracts: Were they applied directly with 50% EtOH content? If so, that might be a bias. Ethanol itself is antibacterial. And the higher added volumes will generate additive effects of ethanol.

Results: What are the significant numbers in the results? It is definitely not the 2-decimals that are used all over in this work. E.g. how did you measure inhibition zones of 8.67±0.47 mm? That is not reliable.

Figure 1 legend: Refer to a table for all these extracts

Line 353-356: Refer to a table showing these extracts.

Discussion: Lines 487-499 + 525-541 + 554-561 + 566-570 + 582-584: This is not discussion of your results!

References: Adequate

Comments on the Quality of English Language

English polishing

Reviewer 2 Report

Comments and Suggestions for Authors

The Reviewer would like to thank the authors for the submission of their article titled, “Vine-winery by-products as precious resource of natural antimicrobials: in vitro antibacterial and antibiofilm activity of grape pomace extracts against foodborne pathogens”.  In this work, the authors hypothesize that polyphenol extracts from wine-production byproducts inhibit the growth of bacterial biofilms.  They address this hypothesis by extracting compounds and performing in vitro assays.  The reviewer will provide comments on both the stylistic components of the paper and the technical aspects.

Style:

1.       The authors show in Figure 1 that extract GpAE7 and GpHE6 exhibited no statistically significant difference in % phenolic content.  Do they have any data for GpAE7 ?  If not, do they plan to investigate this extract further?

2.       Could the authors please provide more references for their suggesting that FICI < 1 indicates synergy?  There are numerous references that would argue that FICI should be less than 0.5 to consider an interaction “synergy”.

3.       Something seems off about Table 1.  The authors report their MIC data as mg/mL.  While this may be correct for their extracts, the MIC for Vancomycin-susceptible SA is, for instance, 2 microgram/mL not 2 miligrams/mL and an MIC of the latter would be highly resistant.  Could they please clarify?

Technical:

1.       Could the authors please clarify, how were the synergy assays conducted?  The reviewer assumes this was through checkerboard or “tube dilution” method but would like to see it explicitly stated in the methods.

2.       For the “Tube Dilution” method, could the authors please describe how they set up the control samples that were used (e.g. without their “extract”).

3.       Could the authors please provide a sample image of their “Zone of Inhibition” experiments?  Adding it to the Supplementary would be fine.

4.       Do the author’s have any data with regards to eradication of existing biofilms (e.g. MBEC).  This might strengthen the paper.

Reviewer 3 Report

Comments and Suggestions for Authors

The research described in the manuscript is devoted to the isolation of extracts from wine production waste and the study of the antibacterial properties of the extracts obtained. The positive aspects of the study include the extraction object available in large quantities. The study was conducted carefully, but there are some questions and comments.

The main remarks.

In my opinion, the novelty of the research is not sufficiently reflected in the article. The discussion does not focus on those points that are new. What are the main achievements of this research? I would highlight the features of the preparation of extracts and the effect of prepared extracts on biofilms. Other aspects of the antibacterial activity of plant extracts of approximately the same composition have been repeatedly described. In addition, it is advisable to discuss possible mechanisms of antibacterial action of extracts and synergistic action with antibiotics of different classes.

Lines 341 - 347 (Results) Phenolic compounds were detected in this study by HPLC. In my opinion, this is not enough. The LC-MS method or some other method should be used to assign peaks on the chromatogram to specific compounds.

Others comments.

Lines 131 - 160 (Section 2.2.) It would be more visual to present the characteristics of the 16 extracts in the form of a table.

Section 3.1. Lines 319 – 324. Concentrations of polyphenols are shown in Fig. 2. Concentration values in the text are superfluous. Authors could provide this data in a table in Supplementary materials.

Section 3.2. What do mg/well or mg/ml concentrations mean when it comes to extracts?

There are some inaccuracies and typos (lines 116-119, 270, 563)

Round 2

Reviewer 1 Report

Comments and Suggestions for Authors

The manuscript has been revised in accordance to several of my previous comments. Some points have not been revised.

The main concern is the use of non-significant numbers. This lowers the quality of the article pretty much, and give the reader an impression that this is not to the point with respect to statistics. Table S1 and S2 should use only significant numbers, and at least one digit (1/100) should be removed from each number.

Comments on the Quality of English Language

Minor language wash. 

Author Response

Table S1 and S2 have been modified as suggested by the reviewer. Changes have been highlighted in Supplementary material.

Reviewer 2 Report

Comments and Suggestions for Authors

The Authors have addressed all of the Reviewer's concerns.

Author Response

Authors thanks Reviewer for providing suggestions to improve the manuscript

Reviewer 3 Report

Comments and Suggestions for Authors

The Authors have made some changes to the text of the manuscript. At least the novelty of the study has become more obvious.

It is somewhat strange that instead of characterizing the available peaks on the chromatograms of the extracts, the Authors decided to completely remove these results.

It is unclear why the Authors inserted a reference to an article about the synergism of antimicrobial peptide arenicin-1 (ref. 56) after the phrase: “A synergism between grape pomace extracts and antibiotics belonging to different classes has been demonstrated [55, 56]”. The presence of antimicrobial peptides in the described extracts has not been mentioned! This reference should be removed.

Author Response

- The results of chemical analysis were removed by the Authors due to requests from other reviewers.

- As suggested by reviewer, reference 56 has been removed in the manuscript and in ‘References’ section. The numbering of references has been adjusted and appropriately highlighted.